# Activity of Estafietin and Analogues on *Trypanosoma cruzi* and *Leishmania braziliensis*

**DOI:** 10.3390/molecules24071209

**Published:** 2019-03-28

**Authors:** Valeria P. Sülsen, Emilio F. Lizarraga, Orlando G. Elso, Natacha Cerny, Andrés Sanchez Alberti, Augusto E. Bivona, Emilio L. Malchiodi, Silvia I. Cazorla, César A. N. Catalán

**Affiliations:** 1CONICET—Universidad de Buenos Aires, Instituto de Química y Metabolismo del Fármaco (IQUIMEFA), Junín 956 2° floor, Buenos Aires 1113, Argentina; orlandoelso@hotmail.com; 2Cátedra de Farmacognosia, Facultad de Farmacia y Bioquímica, Universidad de Buenos Aires, Junín 956 2° floor, Buenos Aires 1113, Argentina; 3Instituto de Fisiología Animal, Fundación Miguel Lillo and Facultad de Ciencias Naturales e Instituto Miguel Lillo, Universidad Nacional de Tucumán, Tucumán T4000INI, Argentina; eflizarraga@lillo.org.ar; 4CONICET—Universidad Nacional de Luján, Instituto de Ecología y Desarrollo Sustentable (INEDES), Ruta 5 y Avenida Constitución, Luján 6700, Argentina; natachacerny@gmail.com; 5Cátedra de Inmunología, Facultad de Farmacia y Bioquímica, Universidad de Buenos Aires, Junín 956 4° floor, Buenos Aires 1113, Argentina; andres.sanchez.alberti@gmail.com (A.S.A.); augustobivona@gmail.com (A.E.B.); emalchio@ffyb.uba.ar (E.L.M.); 6Instituto de Estudios de la Inmunidad Humoral (IDEHU), UBA-CONICET, Junín 956 4° floor, Buenos Aires 1113, Argentina; 7CONICET—Universidad de Buenos Aires, Instituto de Microbiología y Parasitología Médica—CONICET (IMPaM), Facultad de Medicina, Paraguay 2155. 13° floor, Buenos Aires C1121ABG, Argentina; silcazorla@yahoo.es; 8CONICET—Centro de Referencia para Lactobacilos (CERELA), Batalla de Chacabuco 145, San Miguel de Tucumán T4000INI, Argentina; 9CONICET—Universidad Nacional de Tucumán, Instituto de Química del Noroeste—CONICET (INQUINOA), Ayacucho 471, San Miguel de Tucumán T4000INI, Argentina

**Keywords:** sesquiterpene lactones, *Stevia alpina*, *Trypanosoma cruzi*, *Leishmania braziliensis*

## Abstract

Sesquiterpene lactones are naturally occurring compounds mainly found in the Asteraceae family. These types of plant metabolites display a wide range of biological activities, including antiprotozoal activity and are considered interesting structures for drug discovery. Four derivatives were synthesized from estafietin (**1**), isolated from *Stevia alpina* (Asteraceae): 11βH,13-dihydroestafietin (**2**), epoxyestafietin (**3a** and **3b**), 11βH,13-methoxyestafietin, (**4**) and 11βH,13-cianoestafietin. The antiprotozoal activity against *Trypanosoma cruzi* and *Leishmania braziliensis* of these compounds was evaluated. Epoxyestafietin was the most active compound against *T. cruzi* trypomastigotes and amastigotes (IC_50_ values of 18.7 and 2.0 µg/mL, respectively). Estafietin (**1**) and 11βH,13-dihydroestafietin (**2**) were the most active and selective compounds on *L. braziliensis* promastigotes (IC_50_ values of 1.0 and 1.3 μg/mL, respectively). The antiparasitic activity demonstrated by estafietin and some of its derivatives make them promising candidates for the development of effective compounds for the treatment of Chagas disease and leihsmaniasis.

## 1. Introduction

Sesquiterpenelactones (STLs) are a family of naturally occurring compounds mainly found in the Astaeraceae family, though not exclusively. Typically, they are characterized by a C15 mono- or bicyclic skeleton carrying an *exo*-methylene γ-lactone ring. Many biological activities exhibited by STLs have been attributed to the presence of the γ-lactone moiety. Antimicrobial, anti-inflammatory, antiproliferative, and antiprotozoal activities have been reported for these compounds [1,2,3]. Concerning these two last activities, some examples are: artemisinin, from *Artemisia annua*, currently used as antimalarial; psilostachyin, psilostachyin C, cumanin, and deoxymikanolide with trypanocidal activity; and parthenolide, costunolide, helenalin, arglabin, and tapsigargin with antiproliferative activity [4,5,6,7,8]. These findings support continuing with the investigation of this promising group of compounds.

In the beginning, STLs did not arouse much interest because they are frequently highly cytotoxic. However, the possibility of introducing structural modifications to improve its pharmacological properties (efficacy, selectivity, solubility, andtoxicity) has radically modified the panorama and has made its therapeutic application feasible [9,10]. In this sense, simple modifications made either on the lactone ring (opening, dihydrogenation) or on functional groups (OH, COOH) could be a strategy to improve the biological activities or reduce the toxicity [10,11].

According to the World Health Organization (WHO), 6–7 million people are infected with the protozoan parasite *Trypanosoma cruzi*, the causative agent of Chagas disease. *T. cruzi* has a complex life cycle with four distinct developmental forms that alternate between the insect vector (epimastigotes and metacyclic trypomastigotes) and the mammalian host (bloodstream trypomastigotes and amastigotes). This parasitosis has a worldwide distribution, with the highest prevalence values occurring in Latin America. Two drugs are used for the treatment of Chagas disease, i.e., nifurtimox and benznidazole, which have severe adverse effects [12].

Leishmaniasis is another parasitic disease produced by protozoan parasites of the genus *Leishmania*. It is estimated that nearly 1 million new cases and over 20,000 deaths occur annually due to this parasitosis [13]. Leishmaniasis is transmitted by the bite of infected female phlebotomine sandflies. Sandflies inject the infective stage (promastigote) during blood meals. Promastigotes are then phagocytised by macrophages to transform intracellularly into the replicative amastigote, which is the tissue stage of the parasite. *Leishmania (Viannia) braziliensis* has been described as the main causative agent of human tegumentary leishmaniasis in Argentina [14].

Estafietin (**1**) is the main STL produced by *Stevia alpina* (Asteraceae) [15]. Within a series of 15 STLs studied by our research group, estafietin showed significant activity on *T. cruzi* epimastigotes and proved to be selective for the parasite [16]. The trypanocidal activity of this compound was further studied on the infective and intracellular forms of the parasite and estafietin analogues were synthetized.

In this work, estafietin and its dihydro-, epoxy-, methoxy- and ciano- analogues were synthesized and assayed on *T. cruzi* trypomastigotes and amastigotes, as well as on *L. braziliensis* promastigotes. Each compound was also tested for their cytotoxicity on mammalian cells.

## 2. Results

### 2.1. Chemistry

Estafietin (**1**) was isolated from *Stevia alpina* (Asteraceae). The melting point value and the ^1^H- and ^13^C-NMR spectra obtained were identical to those previously reported [15]. From this compound, the derivatives **2**–**5** were synthetized (Figure 1).

The reduction of estafietin (**1**) with sodium borohydride (NaBH_4_) afforded 11βH,13-dihydroestafietin (**2**). Estafietin (**1**) was also subjected to epoxydation with *m*-chloroperbenzoic acid (*m*-CPBA) to afford a 1:4 mixture of α-epoxyestafietin (**3a)** and β-epoxyestafietin (**3b**) respectively as determined by GC and NMR analyses. 11βH,13-methoxyestafietin (**4**) was synthesized by Michael addition of methanol to estafietin (**1**) catalized by sodium hydroxide (NaOH). Similarly, the treatment of estafietin (**1**) with potassium cyanide (KCN) yielded 11βH,13-cianoestafietin (**5**).

### 2.2. Biology

The antiprotozoal activity of estafietin (**1**) and its derivatives (**2**–**5**) were evaluated against *Trypanosoma cruzi* and *Leishmania braziliensis*. The effect of the natural and semisynthetic compounds against *T. cruzi* trypomastigotes is shown in Figure 2.

Epoxyestafietin (**3**) was the most active compound against the infective form of *T. cruzi,* with an IC_50_ value of 18.7 μg/mL, while the natural STL (**1**) showed an IC_50_ value of 25.2 μg/mL. The compounds 11βH,13-dihydroestafietin (**2**) and 11βH,13-cianoestafietin (**5**) were less active (IC_50_ = 97.1 and 78.1 µg/mL, respectively), while 11βH,13-methoxyestafietin (**4**) resulted in being inactive against this parasite stage (IC_50_ > 100 µg/mL). The IC_50_ of the reference drug Benznidazole was 11.6 µg/mL.

The trypanocidal activity of compounds **1**–**5** was also tested on amastigotes, the replicative stage of *T. cruzi* in the human host. Results are shown in Figure 3.

Epoxyestafietin was the most active compound against the intracellular forms, with percentages of inhibition higher than 75% at a concentration of 5.0 μg/mL. This compound presented an IC_50_ value of 2.0 μg/mL. Estafietin (**1**) and 11β,13-methoxyestafietin (**4**) showed similar activity, with IC_50_ values of 28.8 and 30.5 μg/mL, respectively. Compounds **2** and **5** were less active (IC_50_ = 83.0 µg/mL and IC_50_ = 99.0 µg/mL, respectively). The IC_50_ of the reference drug Benznidazole was 1.8 µg/mL.

Since co-infection with *T. cruzi* and *Leishmania* spp. may occur in the North of Argentina [17,18,19], the effect of the compounds on the growth of *Leishmania braziliensis* was also tested. Thus, estafietin (**1**) and 11βH,13-dihydroestafietin (**2**) were the most active compounds, with IC_50_ values of 1.0 and 1.3 μg/mL, respectively. The epoxy derivative (**3**) was less active against this parasite (IC_50_ = 7.8 μg/mL). The compounds 11βH,13-methoxyestafietin (**4**) and 11βH,13-cianoestafietin (**5**) showed moderate activity, with IC_50_ values of 48.7 and 51.8 μg/mL, respectively (Figure 4). Amphotericin B showed an IC_50_ of 0.48 µg/mL.

The cytotoxicity of the natural and synthetic compounds was evaluated on Vero cells by the MTT method. The 50% cytotoxic concentration (CC_50_) of each compound was determined. Selectivity indexes (SI) were calculated for *T. cruzi* trypomastigotes and amastigotes and for *L. braziliensis* promastigotes (Table 1).

Estafietin (**1**) and estafietin derivatives **2**, **4** and **5** showed low toxicity on mammalian cells. Estafietin presented a CC_50_ value of 240.4 µg/mL, with SI of 9.5, 8.3 and 240.4 for *T. cruzi* trypomastigotes and amastigotes and *L. braziliensis* promastigotes, respectively. CC_50_ values for compounds **2** and **4** were 164.6 and 315.6 µg/mL, respectively. Selectivity indexes for these derivatives were 126.6 and 10.3 for promastigotes and amastigotes, respectively. The CC_50_ value for compound **3** was 8.1 µg/mL and the SIs for this STL were lower than those obtained for other derivatives.

## 3. Discussion

Analogues of estafietin were synthesised with the aim of comparatively evaluating the anti-protozoal activity of the natural compound and its derivatives. The strategy used to obtain estafietin derivatives was to make modifications on the functionalizations most frequently found in STLs. In this sense, considering the reactivity of the epoxide moiety, we decided to synthesise the 10(14)-epoxy derivative of estafietin. A 1:4 mixture of α- and β-epoxides was obtained as it was described by De Heluani et al. [15]. The selective hydrogenation of the C11-C13 double bond was performed with sodium borohydride to obtain 11βH,13-dihydroestafietin. The two other derivatives, 11βH,13-methoxyestafietin (**4**) and 11βH,13-cianoestafietin (**5**) were obtained by a Michael type addition at the C11–C13 double bond employing sodium methoxide and sodium cyanide.

Estafietin and its derivatives were assayed on *T. cruzi* trypomastigotes and amastigotes and on *L. braziliensis* promastigotes. The epoxy derivative of estafietin was more active than the natural compound against the infective form of *T. cruzi* (IC_50_ = 18.7 µg/mL). Sesquiterpene lactones that have saturated the C11–C13 exocyclic double bond were either less active (compounds **2** and **4**) or inactive (compound **5**). When the STLs were evaluated against amastigotes, epoxiestafietin also showed the greatest activity (IC_50_ = 2.0 µg/mL), whereas 11βH,13-methoxyestafietin (**4**) presented activity values that were similar to those obtained with the natural compound estafietin (**1**).

A different behavior was observed when the anti-leishmanial activity was evaluated. In this case, estafietin (**1**) was the most active compound with an IC_50_ =1.0 µg/mL. Among the derivatives, 11βH,13-dihydroestafietin was the most active (IC_50_ = 1.3 µg/mL), followed by epoxyestafietin (IC_50_ = 7.8 µg/mL).

The cytototoxicity of estafietin and its derivatives was evaluated on Vero cells. Estafietin (**1**) exhibited selectivity against the infective and intracellular forms of *T. cruzi* (SI = 9.5 and 8.3, respectively), while 11βH,13-methoxyestafietin (**4**) was not toxic for mammalian cells at the concentration active against *T. cruzi* amastigotes. Estafietin (**1**) and 11βH,13-dihydroestafietin (**2**) were selective against *L. braziliensis* promastigotes (SI = 240.4 and 126.6, respectively). In spite of being active against *T. cruzi* and *L. braziliensis*, epoxyestafietin (**3**) has not shown selectivity of action as antiprotozoal.

Most STLs have an exocyclic methylene group conjugated to the carbonyl of the γ-lactone moiety. Many of the biological activities of STLs are attributed to the presence of this functionality. The absence of this α,β-unsaturated carbonyl group would render the sesquiterpene lactone inactive. However, many sesquiterpene lactones having this group reduced, such as artemisinin, matricin and santonin, are highly active [1,20]. This phenomenon is in agreement with the activity displayed by 11βH,13-dihydroestafietin against *L. braziliensis*.

The results obtained show that simple modifications on the sesquiterpene lactones structure might render in most active and selective compounds. Other estafietin analogues can be further synthesised aiming to enhance the antiprotozoal activity and selectivity of action.

## 4. Materials and Methods

### 4.1. Plant Material

The aerial parts of *Stevia alpina* Griseb. (Asteraceae) were collected at the flowering stage during the first fortnight of March 2011 at km 45 of route 307 to Tafí del Valle, Tucumán province, Argentina. The plant material was left to dry in the shade for 2 weeks at room temperature. A voucher specimen was deposited at the herbarium of the Instituto Miguel Lillo, S. M. de Tucumán (Tucumán, Argentina).

### 4.2. Isolation of Estafietin *(**1**)*

Estafietin (**1**) was isolated from *Stevia alpina* according to the procedure described by Mercado et al. [21] with some modifications. The air-dried leaves and flowers (340 g) were placed in an Erlenmeyer flask (6 L) and 3.8 L of dichloromethane was quickly added at room temperature with a continuous and gentle swirling movement for 40–60 s. The mixture was then filtered through filter paper and the solvent evaporated at reduced pressure to afford 33.5 g (9.8% yield) of crude extract which was suspended in ethanol 96° (300 mL) at ca. 65°, diluted with water (160 mL) and extracted successively with hexane (200 mL, 150 and 100 mL) and then with dichloromethane (3 × 300 mL). The evaporation of the hexane fraction rendered 20.1 g of residue which was discarded. The evaporation of the dichloromethane extract at reduced pressure furnished a residue (9.7 g) which was dissolved in the minimum amount of ethyl ether and stored over the weekend in the refrigerator. From this solution, somewhat impure estafietin (0.58 g) precipitated. The supernatant was taken to dryness and the residue (8.1 g) was subjected to column chromatography on Silicagel (70–230 mesh; 250 g) using chloroform with increasing amounts of Et_2_O (0–15%) as mobile phase. Seventy-eight fractions were collected and monitored by TLC. Fractions, 37–51 which showed a major spot on TLC, were pooled and the solvent evaporated to obtain a crystalline residue (2.8 g) of impure estafietin. Both impure estafietin precipitates were joined and recrystallized from heptane-ethyl acetate affording 1.52 g of a pure lactone, with an mp 102–104 °C and whose ^1^H- and ^13^C-NMR spectra (see Table 2 and Table 3) matched with those reported previously [14]. EIMS (70 eV) *m*/*z* (rel. int.): 246 ([M]^+^, 13), 231 (54), 228 (8), 218 (6), 213 (5)203 (11), 190 (9), 187 (12), 175 (18), 162 (24), 149 (25),131 (28), 117 (31), 105 (43), 97 (100), 91 (88), 79 (58), 77 (60), 67 (38), 65 (31), 53 (83), 43 (90), 41 (67). The NMR spectra are available as Appendix A.

### 4.3. Synthesis of Estafietin Derivatives

#### 4.3.1. 11βH,13-Dihydroestafietin (**2**)

The stereoselective reduction of the 11(13)-exocyclic double bond of estafietin (**1**) was carried out using sodium borohydride (NaBH_4_). To a solution of estafietin (**1**) (200 mg; 0.813 mmol) in methanol (7 mL) was added dropwise with stirring a solution of sodium borohydride (90 mg) in 2.5 mL of methanol at 0 °C. After 45 min, the solution was acidified with a 10% aqueous solution of acetic acid (5 mL), diluted with 15 mL of water and extracted with chloroform (3 × 40 mL). The organic layer was washed with a 5% aqueous solution of sodium bicarbonate (2 × 4 mL) and water and was then dried with anhydrous sodium sulfate and the solvent evaporated at reduced pressure. The residue (145 mg) was subjected to column chromatography on Silicagel to yield 105 mg (52% yield) of pure compound **2** as fine needles. EIMS (70 eV) *m*/*z* (rel. int.) : 248 ([M]^+^, 12), 233 (55), 230 (10), 220 (5), 215 (5), 205 (9), 192 (7), 190 (7), 177 (21), 164 (16), 152 (53), 151 (52), 131 (34), 123 (31), 107 (28), 105 (44), 97 (100),95 (62), 91 (79), 79 (68), 77 (58), 69 (38), 67 (46), 55 (86),53 (43), 43 (87), 41 (80). ^1^H- and ^13^C-NMR in Table 2 and Table 3, respectively. The NMR spectra are available as Appendix A.

#### 4.3.2. α-10(14)-epoxyestafietin (**3a**) and β-10(14)-epoxyestafietin (**3b**)

To estafietin (200 mg; 0.813 mmol) in dichloromethane (12 mL) and 2.5 mL of 0.5 M sodium bicarbonate cooled in ice, was added in small portions 450 mg of *m*-chloroperbenzoic acid with magnetic stirring. The reaction mixture was left overnight with stirring, diluted with chloroform (20 mL), and the organic layer was washed with 10% sodium thiosulfate (3 × 3 mL), 1M NaOH (2 × 3 mL), water (3 × 3 mL), and dried with anhidrous sodium sulfate and taken to dryness in a rotary evaporator. The residue was subjected to column chromatography on Silicagel (230–400 mesh) with hexane-ethyl acetate mixtures of increasing polarity as a mobile phase to give 51 mg of unchanged estafietin and 69 mg of crystals consisting of a 1:4 mixture of α- and β-epoxides **3a** and **3b** respectively as determined by GC and NMR analysis. EIMS (70 eV) *m*/*z* (rel. int.) of 10(14)-α-epoxide **3a** by GC-MS : 262 ([M]^+^, 1.4), 247 (20), 244 (6), 233 (13),219 (8), 215 (8), 204 (2), 201 (8), 191 (5), 187 (8), 173 (13), 162 (8), 159 (12), 145 (20), 131 (18), 119 (23), 117 (25), 105 (31), 95 (56), 91 (63), 86 (18), 84 (29), 81 (36), 79 (50), 77 (44), 69 (19), 67 (32), 65 (26), 55 (38),53 (59), 43 (100), 41 (64). EIMS (70 eV) *m*/*z* (rel. int.) of 10(14)-β-epoxide **3b** by GC-MS : 262 ([M]^+^,3), 247 (4), 244 (5), 233 (34),219 (4), 215 (7), 204 (4), 201 (5), 191 (7), 187 (8), 173 (15), 162 (11), 159 (11), 145 (28), 131 (26), 119 (25), 117 (27), 105 (28), 95 (31), 91 (57), 86 (8), 84 (7), 81 (30), 79 (43), 77 (37), 69 (15), 67 (26), 65 (22), 55 (29),53 (53), 43 (100), 41 (52). The ^1^H-NMR spectrum of the mixture was in complete agreement with the data reported for the pure epoxides [14]. The previously unreported ^13^C-NMR of β-epoxide **3b** (125 MHz in CDCl_3_): δ 169.5 ppm (C-12), 139.2 (C-11), 120.2 (C-13), 79.9 (C-6), 65.7 (C-4), 62.8 (C-3), 57.1 (C-10), 53.1 (C-14), 50.3 (C-5), 47.3 (C-7), 42.5 (C-1), 28.5 (C-2), 24.2 (C-8 and C-9), 19.0 (C-15); α-epoxide **3a**: δ 169.7 ppm (C-12), 139.5 (C-11), 120.8 (C-13), 80.2 (C-6), 65.9 (C-4), 62.9 (C-3), 57.5 (C-10), 56.7 (C-14), 49,8 (C-5), 43.4 (C-7), 42.0 (C-1), 18.4 (C-15). The NMR spectra are available as Appendix A.

#### 4.3.3. 11βH,13-Methoxyestafietin (**4**)

To a solution of **1** (100 mg, 0.406 mmol) in anhydrous methanol (10 mL), 1.0 M NaOH in methanol (1 mL) was added, left at room temperature for 10 min, and then a 10% acetic acid solution was added dropwise to reach a pH of 5–6. After evaporating the solvent, the residue was subjected to column chromatography on Silicagel (230–400 mesh) using hexane-ethyl acetate mixtures of increasing polarity as a mobile phase to give lactone **4** (66 mg, 68% yield) as an amorphous white powder. EIMS (70 eV) *m*/*z* (rel. int.): 278 (M^+^, 4), 263 (19), 260 (2), 248 (5), 233 (8), 231 (12), 215 (7), 182 (13), 175 (15), 149 (15), 131 (18), 117 (16), 105 (21), 97 (6), 91 (40), 79 (29), 77 (25), 55 (24), 53 (19), 45 (100), 43 (42), 41 (35). ^1^H- and ^13^C-NMR in Table 2 and Table 3, respectively. The NMR spectra are available as Appendix A.

#### 4.3.4. 11βH,13-Cianoestafietin (**5**)

To a solution of **1** (100 mg, 0.406 mmol) in anhydrous methanol (10 mL), 300 mg of potassium cyanide was added, the mixture was heated to reflux, and then 4.0 mL of 0.1 N acetic acid in methanol was added dropwise in 2 hours. The solution was further refluxed for 3 hours, poured into cold water (90 mL) and extracted with chloroform (3 × 50 mL). The chloroform extract was dried with anhydrous sodium sulfate, filtered and brought to dryness in a rotary evaporator to give a yellowish oil residue which was subjected to column chromatography on Silicagel (230–400 mesh) employing hexane-ethyl acetate mixtures of increasing polarity as mobile phase to give lactone **5** (59 mg, 53% yield) as a colorless gum. EIMS (70 eV) *m*/*z* (rel. int.): 273 (M^+^, 8), 258 (29), 255 (9), 230 (10), 215 (13), 202 (7), 189 (11), 176 (22), 131 (28), 117 (25), 105 (32), 97 (89), 91 (67), 84 (24), 80 (31), 79 (57), 77 (47), 69 (21), 67 (32), 65 (23), 55 (30), 53 (52), 49 (25), 43 (100), 41 (66). ^1^H- and ^13^C-NMR in Table 2 and Table 3, respectively. The NMR spectra are available as Appendix A.

### 4.4. Parasites

*Trypanosoma cruzi* epimastigotes (RA strain) were grown in a biphasic medium. Cultures were routinely maintained by weekly passages at 28 °C. *T. cruzi* bloodstream trypomastigotes expressing β-galactosidase (Tul-β-Gal) were obtained from infected CF1 mice by cardiac puncture at the peak of parasitemia on day 15 post-infection [22]. Trypomastigotes were routinely maintained by infecting 21-day-old CF1 mice (2943/2013, Review Board of Ethics of Faculty of Medicine, University of Buenos Aires, Argentina).

*Leishmania braziliensis* promastigotes (MHOM/BR/75/M2903 strain) were grown in liver infusion tryptose medium (LIT). Cultures were routinely maintained by weekly passages at 26 °C. Parasites were passaged 24 or 48 h before to the experiments.

### 4.5. In Vitro Trypanocidal Activity

The trypanocidal effect of the purified compounds was tested on transfected bloodstream trypomastigotes (Tulhauen strain). Briefly, mouse blood containing trypomastigotes was diluted in complete LIT medium to adjust the parasite concentration to 2 × 10^6^/mL. Parasites were seeded in duplicate in a 96-well microplate in the presence of each compound (0–100 µg/mL) and controls and incubated at 4 °C for 24 h. The number of remaining living parasites in each sample was determined in 5 µL of cell suspension diluted 1/5 in lysis buffer (0.75% NH_4_Cl, 0.2% Tris, pH 7.2) and counted in a Neubauer chamber.

The effect of the compounds on intracellular forms of *T. cruzi* was assayed using β-galactosidase transfected parasites [6]. Ninety-six well culture plates were seeded with nonphagocytic Vero cells at 5 × 10^3^ per well in 100 mL of culture medium and incubated for 24 h at 37 °C in a 5% CO_2_ atmosphere. Cells were then washed and infected with transfected bloodstream trypomastigotes expressing β-galactosidase at a parasite/cell ratio of 10:1. After 12 h of coculture, plates were washed twice with PBS to remove unbound parasites and each pure compound was added at different concentrations (0–100 µg/mL) in 150 µL of fresh complete RPMI medium without phenol red (Gibco, Rockville, MD, USA). Controls included infected untreated cells (100% infection control) and uninfected cells (0% infection control). The assay was developed by the addition of chlorophenolred-β-d-galactopyranoside (CPRG) (100 mM) and 1% Nonidet P40, 5 days later. Plates were incubated for 4–6 h at 37 °C and the absorbance was measured at 595 nm in a microplate reader (Bio-Rad Laboratories, Hercules, CA, USA). The percentage inhibition was calculated as 100 – {[(absorbance of treated infected cells)/(absorbance of untreated infected cells) × 100]} and the IC_50_ value was estimated. Benznidazole was used as positive control.

### 4.6. In Vitro Leishmanicidal Activity

The growth inhibition of *Leishmania braziliensis* promastigotes was evaluated by the (^3^H) thymidine [(^3^H)T] uptake assay. Parasites (5 × 10^6^) were settled at a final volume of 150 µL in a flat-bottom 96-well microplate and cultured at 28 °C in either the absence or the increasing concentrations of the pure compounds (0–50 µg/mL). After 72 h, 1 µC of [^3^H]T was added to each well, and cultures were incubated for 16 h. Counts per minute (cpm) of triplicate cultures were measured by an automated liquid scintillation counter (Packard, Downers Grove, IL, USA). The percentage of inhibition was calculated as {100 − [(cpm of treated parasites/cpm of untreated parasites) × 100]}. The compound concentration at which the parasite growth was inhibited by 50% (IC_50_) was determined. Amphotericin B was used as positive control.

### 4.7. Host Cell Toxicity

Vero cells were employed to determine the viability by the 3-(4,5-dimethylthiazol-2yl)-2,5-diphenyltetrazolium bromide (MTT) method [6]. Cells (5 × 10^5^) were settled at a final volume of 150 mL in a flat-bottom 96-well microplate and cultured at 37 °C in a 5% CO_2_ atmosphere in either the absence or the presence of increasing concentrations of the compounds (5–500 µg/mL). After 24 h, MTT was added at a final concentration of 1.5 mg/mL. Plates were incubated for 2 h at 37 °C. The purple formazan crystals formed were dissolved with 150 mL of ethanol and the absorbance was read at 570 nm in a microplate reader. Results were calculated as the ratio between the optical density in the presence and absence of the compound multiplied by 100.

### 4.8. Statistical Analysis

Results are presented as means ± SEM. The GraphPad Prism 5.0 software (GraphPad Software Inc., San Diego, CA, USA) was employed to carry out calculations. To calculate IC_50_ and CC_50_ values, the percentages of inhibition were plotted against the drug concentration and fitted with a straight line determined by a linear regression (Sigma Plot 10 software, (Systat Software, San Jose, CA, USA). Results presented are representative of three to four independent experiments.

The statistical significance was determined by the Kruskal-Wallis test performed with the GraphPad Prism 5.0 software (GraphPad Software Inc., San Diego, CA, USA). Each treatment was compared to controls. *p* values < 0.05 were considered significant.

## 5. Conclusions

Sesquiterpene lactones are considered privileged structures that can be taken as a starting point for the search and development of new drugs due to their lipophilicity, the presence of Michael acceptor groups and certain steric and electronic characteristics. The antiparasitic activity demonstrated by estafietin and some of its derivatives, in addition to these other characteristics, makes them promising candidates for future studies.

## Figures and Tables

**Figure 1 molecules-24-01209-f001:**
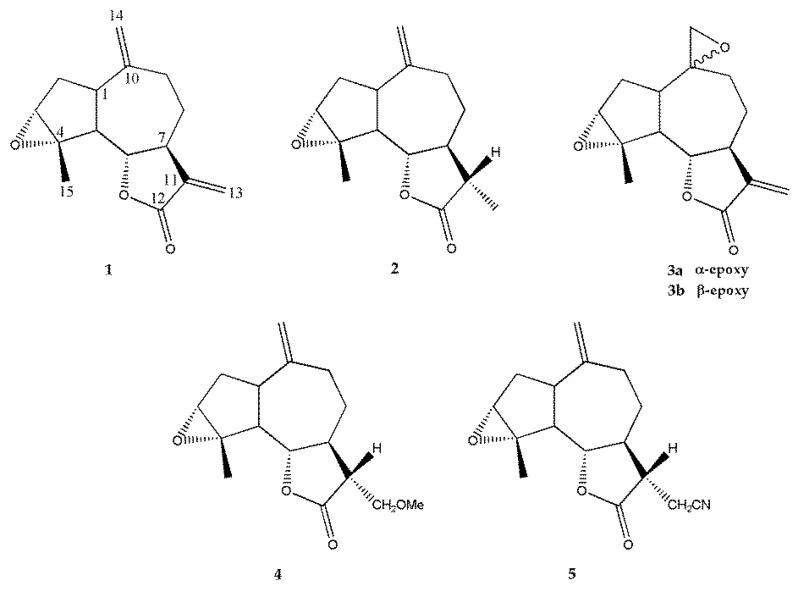
Chemical structures of estafietin (**1**) and its derivatives (**2–5**).

**Figure 2 molecules-24-01209-f002:**
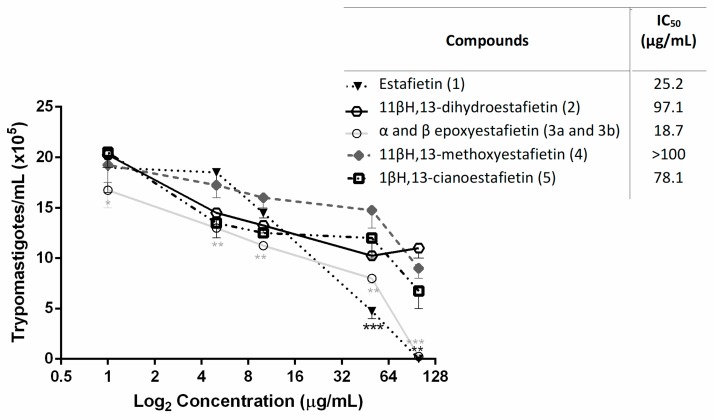
Trypanocidal activity of estafietin (**1**) and estafietin derivatives (**2**–**5**) against *T. cruzi* bloodstream trypomastigotes. Bloodstream trypomastigotes were assayed in duplicate in the presence of 0 to 100 µg/mL of estafietin (**1**), 11βH,13-dihydroestafietin (**2**), α- (**3a**) and β-epoxyestafietin (**3b**) (1:4 mixture), 11βH,13-methoxyestafietin (**4**) and 11βH,13-cianoestafietin (**5**). Cultures were done in 96-well plates with 1.5 × 10^6^ parasites/mL over 24 h. The living parasites were then counted in a Neubauer chamber. Symbols represent mean ± SEM. Each treatment was compared to controls: parasites incubated with culture medium only. * *p* < 0.05, ** *p* < 0.01 and *** *p* < 0.005.

**Figure 3 molecules-24-01209-f003:**
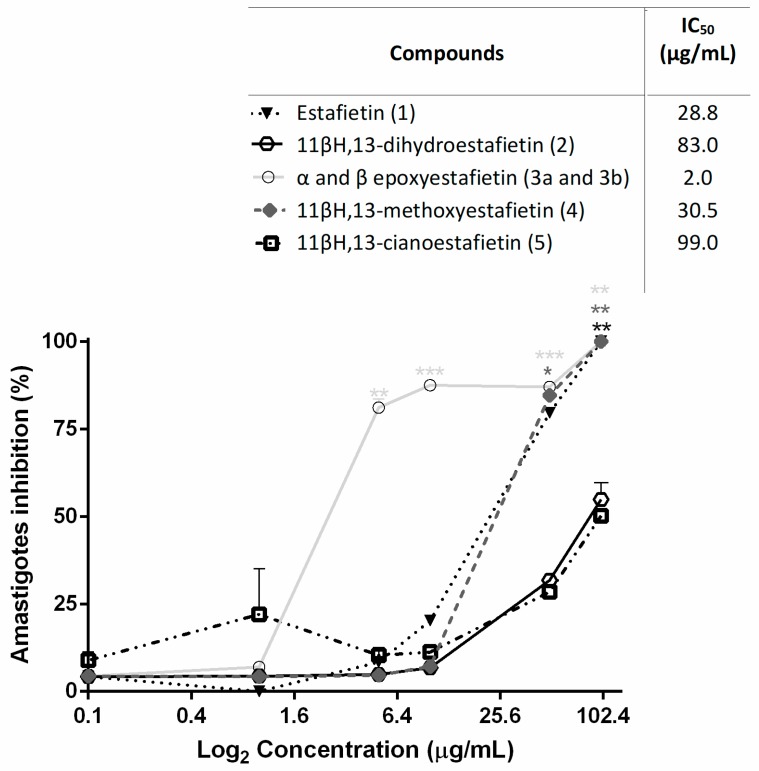
Inhibition of *T. cruzi* amastigotes replication by estafietin (**1**) and estafietin derivatives (**2**–**5**). Mammalian cells (5 × 10^3^ cells/well) were seeded in 96-well plates and infected 24 h later with transfected trypomastigotes expressing β-galactosidase. After 24 h of co-culture, plates were washed and compounds were added at 0–100 µg/mL in 150 µL medium. On day 6 post-infection, the assays were developed by the addition of CPRG (100 mM) and Nonidet P-40 (1%). Plates were incubated for 6 h and the optical density was read at 570 nm. Infected untreated mammalian cells were used as 100% infection control. The percentage of inhibition was calculated as 100 − {[(Absorbance of treated infected cells)/(Absorbance of untreated infected cells)] × 100}. Symbols represent mean ± SEM. Each treatment was compared to controls: parasites incubated with culture medium only. * *p* < 0.05, ** *p* < 0.01 and *** *p* < 0.005.

**Figure 4 molecules-24-01209-f004:**
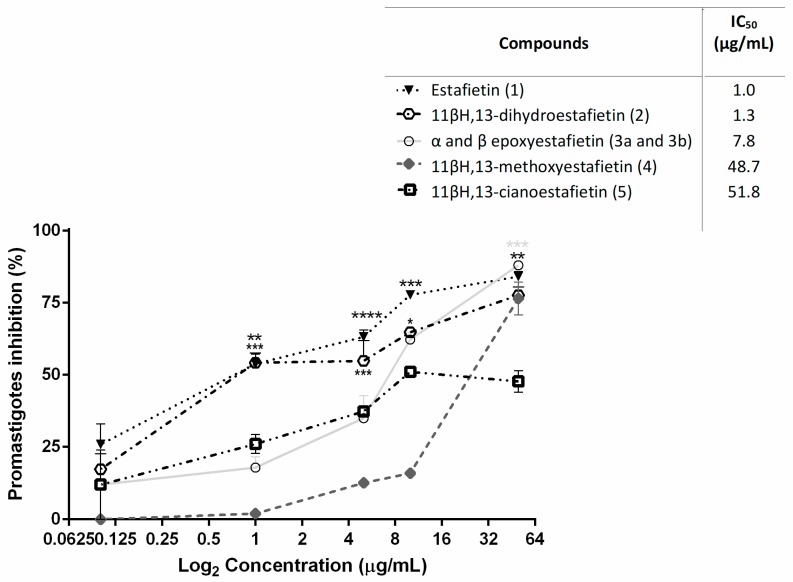
Leishmanicidal activity of estafietin (**1**) and estafietin derivatives (**2**–**5**) on *L. braziliensis* promastigotes. Parasites were cultured for 72 h in the presence of each compound (0–50 µg/mL). The inhibition of parasite growth was evaluated by a [^3^H] thymidine uptake assay. The percentage of inhibition was calculated as 100 − [(cpm of treated parasites)/(cpm of untreated parasites)] × 100. Values represent mean ± SEM from three independent experiments carried out in triplicate. Each treatment was compared to controls: parasites incubated with culture medium only. * *p* < 0.05, ** *p* < 0.01 and *** *p* < 0.005.

**Table 1 molecules-24-01209-t001:** Cytotoxicity on Vero cells and selectivity indexes of estafietin (**1**) and estafietin derivatives (**2**–**5**).

Compounds	CC_50_ (µg/mL)	Selectivity Index		
		Trypomastigotes *T. cruzi*	Amastigotes *T. cruzi*	Promastigotes *L. braziliensis*
**1**	240.4	9.5	8.3	240.4
**2**	164.6	1.7	2.0	126.6
**3**	8.1	0.4	4.0	1.0
**4**	315.6	3.1	10.3	6.1
**5**	133.6	1.7	1.3	2.7

**Table 2 molecules-24-01209-t002:** ^1^H-NMR data of compounds **1**–**5** in CDCl_3_ (600 MHz; TMS as internal standard).

H	1	2	3b	4	5
	δ (J in Hz)	δ (J in Hz)	δ (J in Hz)	δ (J in Hz)	δ (J in Hz)
**1**	2.98 ddd	2.90 ddd	2.22 ddd	2.91 ddd	2.93 ddd
	(10.5, 8.5, 7.5)	(10.5, 8.5, 7.3)	(10.5, 8.3, 8)	(10.5, 8.5, 7.3)	(10.5, 8, 7.3)
**2a**	2.07 dd	2.10 dd	2.10 dd	2.10 dd	2.15 dd
	(14, 7.5)	(14, 7)	(13.8, 8)	(13.9, 7.3)	(13.9, 7.3)
**2b**	1.81 ddd	1.80 ddd	1.80 ddd	1.80 ddd	1.80 ddd
	(14, 10.5, 1.2)	(14, 10.5, 1.2)	(13.8, 10.5, 1.5)	(13.9, 10.5, 1.2)	(13.9, 10.5, 0.7)
**3**	3.38 s br	3.36 s br	3.36 s br	3.36 s br	3.37 s br
**5**	2.32 dd	2.28 dd	2.43 dd	2.32 dd	2.35 dd
	(11, 8.5)	(10.7, 8.4)	(11, 8.3)	(10.7, 8.5)	(10.7, 8)
**6**	4.08 dd	3.96 dd	4.12 dd	3.99 dd	4.09 dd
	(11, 8.8)	(10.5, 9.7)	(11, 9)	(10.7, 9.4)	(10.7, 9.5)
**7**	2.87 ddddd	1.91 m	2.83 m	2.36 m	2.27 m
	(11.5, 8.8, 5.2, 3.6, 3.2)				
**8a**	2.22 m	2.09 m	2.10 m	2.17 m	2.34 m
**8b**	1.53m	1.35 m	1.60 m	1.38 m	1.51 m
**9a**	2.28 m	2.29 m	2.00 m	2.26 m	2.32 m
**9b**	2.19 m	2.12 m	1.60m	2.14 m	2.20m
**11**	---	2.22 dq	---	2.41ddd	2.58 ddd
		(12, 7)		(11.7, 4.7, 3.4)	(11.8, 7.8, 4.4)
**13a**	6.21 d	1.22 ^ʘ^ d (7)	6.23 d	3.66 * dd	2.87 * dd
	(3.6)		(3.5)	(9.7, 4.7)	(17.2, 4.4)
**13b**	5.48 d		5.51 d	3.63 * dd	2.63 * dd
	(3.2)		(3.1)	(9.7, 3.4)	(17.2, 7.8)
**14a**	4.95 s br	4.88 s br	2.64 * d	4.89 s br	4.93 s br
			(4.8)		
**14b**	4.86 d	4.83 s br	2.61 * d	4.84 s br	4.88 s br
	(1.7)		(4.8)		
**15 ^ʘ^**	1.62 s	1.59 s	1.63 s	1.58 s	1.58 s
**Others**	---	---	---	3.36 s (OMe)	---

^ʘ^ Intensity three protons; * AB system; s = singlet; d = doublet; t = triplet; q = quartet; m = multiplet; br = broad.

**Table 3 molecules-24-01209-t003:** ^13^C-NMR data of compounds **1**–**5** in CDCl_3_ (600 MHz; TMS as internal standard).

C	1	2	3b (*)	4	5
	δ	δ	δ	δ	δ
**1**	44.9 d	44.2 d	42.5 d	44.3 d	44.2 d
**2**	33.0 t	32.7 t	28.5 t	32.8 t	32.6 t
**3**	63.2 d	63.1 d	62.8 d	63.2 d	63.0 d
**4**	65.8 s	66.0 s	65.7 s	66.1 s	65.8 s
**5**	50.8 d	50.4 d	50.3 d	50.3 d	50.2 d
**6**	80.5 d	80.8 d	79.9 d	80.8 d	81.1 d
**7**	44.1 d	50.0 d	47.3 d	44.7 d	47.5 d
**8**	29.2 t	30.9 t	24.2 t	31.2 t	31.0 t
**9**	28.6 t	31.0 t	24.2 t	30.9 t	30.4 t
**10**	146.1 s	147.3 s	57.1 s	147.2 s	146.3 s
**11**	139.6 s	41.9 d	139.2 s	47.9 d	43.6 d
**12**	169.7 s	178.4 s	169.5 s	175.8 s	174.1 s
**13**	120.2 t	13.2 q	120.2 t	69.2 t	17.1 t
**14**	115.3 t	114.1 t	53.1 t	114.2 t	114.9 t
**15**	18.5 q	18.7 q	19.0 q	18.7 q	18.6 q
**Others**	---	---	---	53.9 (q; OMe)	116.7 (s; CN)

(*): 125 MHz.

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
