# Peer review of "Activity of Estafietin and Analogues on Trypanosoma cruzi and Leishmania braziliensis"

_molecules, 2019, doi:10.3390/molecules24071209_

Round 1

Reviewer 1 Report

The manuscript descibes the activity of the natural sesquiterpenoid estafietin and some synthetic analogues agaist different parasites, thus adding new knowledge concernig the biological activity of this class of compounds. The experimental work seems to be state of the art, nevertheless, the activity of positive controls is missing, which is inevitable to judge the relevance of the results. These values have to be added.

Some further points should be considered:

page 3, line 100: GC and NMR analyses

page 3, line 101: estafiein has to be changed to estafietin

page 4, line 119: Living parasites

page 6, line 222: higly has to be changed to highly

page 7, line 245: ethanol 96% ?

page 9, line 293: 125 MHz for the 13C NMR

page 12: references 3 and 5, please add the title

Author Response

We have revised and corrected our paper “Activity of estafietin and analogues on Trypanosoma cruzi and Leishmania braziliensis according to reviewers’ suggestions. Modifications have been highlighted.

1)      Activity of the positive controls has been included as supplementary material

2)      Page 3, line 100: “GC and NMR analyse” has been replaced by “GC and NMR analyses”

3)      Page 3, line 101: “estafiein” has been replaced by “estafietin”

4)   Page 4, line 119: “The live parasites” has been replaced by “The living parasites”

5)      Page 6, line 222: “higly” has been replaced by “highly”

6)      Page 7, line 245: ethanol is expressed in grades (grade alcohol)

7)      Page 9, line 239: 125 MHz for the 13C NMR has been indicated for compound 3b

8)      Page 12: titles of references 3 and 5 have been included

Reviewer 2 Report

I have read with interest the manuscript “Activity of estafietin and analogues on Trypanosoma cruzi and Leishmania braziliensis”

This is an interesting and well written manuscript, however prior to further processing of the paper several points need to be clarified

-In Results section

1.      It is difficult to understand the Fig 2 and 3. In my opinion, a table with IC50 of compounds instead would be of great help when comparing the activity of compounds and the interpretation of results.

2.      Although Leishmania and trypanosome are kinetoplastids, their biological behavior and their sensitivity to drugs is different, so it is difficult to find an effective compound for both parasites. So I find compound 4 more interesting, which also has a good selectivity index in amastigotes, I suggest the authors performe  in vivo studies to assess the efficacy of this compound in the future

Author Response

We have revised and corrected our paper “Activity of estafietin and analogues on Trypanosoma cruzi and Leishmania braziliensis according to reviewers’ suggestions. Modifications have been highlighted.

1)      Tables with the IC50 values of the compounds have been included in the manuscript.

2)      We thank the reviewer suggestion. We will continue with the study of these compounds and we will perform in vivo studies in the future.

Round 2

Reviewer 1 Report

The manuscript might now be accepted.